# Learning Under Multi-dimensional Domain Shifts: A Ensemble of Mixtures of Experts Approach

## Abstract

Domain shifts pose a significant challenge in deep learning applications. Existing methods typically address domain shifts by treating each domain in isolation, overlooking the underlying factors driving the shifts, or focus on only *one* factor. However, domain shifts in the real world often occur across *multiple* dimensions simultaneously. For example, medical datasets from different hospitals can exhibit variations in factors including demographics, equipment manufacturers, and imaging protocols, demonstrating a three-dimensional shifts. In this paper, we introduce a novel approach to address the complexity of multi-dimensional domain shifts. Our method leverages an ensemble of mixtures of experts (EMoE), with each MoE specialized in different dimensions. Crucially, we innovate a domain estimator to address a particularly challenging issue frequently encountered in practice: domain labels may be missing or unreliable. A significant advantage of our method is its generalizability and adaptability to both centralized and federated learning settings, as well as its versatility across various tasks. Extensive experiments on six datasets demonstrate the superiority of our method over state-of-the-art domain generalization and personalized federated learning approaches.

## 1 Introduction

Deep learning has significantly advanced computer vision tasks in recent years. However, deploying these models in real-world scenarios often encounters severe performance degradation due to the problem of domain shifts (Quinonero-Candela et al., 2008). Domain Generalization (DG) (Wang et al., 2022) has been proposed to address these challenges, focusing on developing models robust to changes in data distributions. A promising direction within these efforts is to characterize data of data distributions as mixtures of domains, training models on one mixture and then aiming to generalize across unknown mixtures, involving both seen and unseen domains.

Traditional DG studies typically use domain labels indexed by a single factor. In contrast, real-world domain shifts often emerge from multiple factors. For instance, in medical image analysis, shifts can be attributed to differences in ethnic groups, equipment manufacturers, or imaging protocols and workflows (Karani et al., 2018; Ciompi et al., 2017; Garrucho et al., 2022; Mårtensson et al., 2020). Without utilizing the structure of multi-dimensional domains, existing methods fail to adapt to the shifts across different domain dimensions simultaneously. Thus, it is crucial to consider domain dimensions concurrently as they convey various aspects of data. For example, in medical image analysis, domain shifts from different manufacturers or imaging protocols can affect model robustness, while shifts among different ethnic groups of patients raise significant concerns about fairness, as discussed in Parikh et al. (2019); Seyyed-Kalantari et al. (2020); Hinton (2018).

Due to the inherent heterogeneity of the data sources, domain shifts become even more challenging in federated learning (FL), where models are trained across multiple decentralized clients without exchanging local data samples. Personalized Federated Learning (PFL) (Sattler et al., 2020) and Federated Domain Generalization (FDG) (Liu et al., 2021) were proposed to mitigate this problem and improve generalization on clients involved in FL and unseen datasets outside the federated training, respectively. Yet, these works presuppose that clients each have distinct and isolated domains.

Figure 1: Illustration of multi-dimensional domain shifts in centralized and federated learning settings, featuring an example of 2-dimensional indexed domains based on color and shape. Different icons represent samples from different domains.

This assumption does not hold in many real-world scenarios, such as in medical images, where the data source at a single client may contain a variety of devices or diverse patient demographics.

In this paper, we address the complex issue of domain shifts in both centralized and federated learning scenarios, where data distributions are a mixture of multi-dimensionally indexed domains. Our motivation stems from the need to effectively generalize across unseen mixtures of data domains, as illustrated in Figure 1. A straightforward approach would be to group data with the same domain indices across all dimensions and apply existing DG techniques. However, this method is fundamentally flawed, as certain groups may be underrepresented or entirely absent during training due to the granular domain labels, making it difficult to ensure generalizability. Moreover, this approach overlooks the inherent structure of domain labels, leading to suboptimal model performance.

To address these challenges, we introduce the Ensemble of Mixtures of Experts (EMoE) and its federated learning counterpart, FedEMoE. For each domain dimension, we design a Mixture of Experts (MoE) to address specific domain shifts, and we ensemble these MoEs to account for all potential domain shifts factors. Our model is explicitly optimized to ensure equal effectiveness across all domains within each dimension, enabling it to generalize robustly across a wide range of test set mixtures. This design maximally leverages information from structured domain labels.

A significant challenge arises when domain labels are partially missing, or when predefined domain labels cannot accurately characterize the data distribution. This is common in practical scenarios, where domain labels—like patient demographics or equipment manufacturers—may be incomplete or unable to fully capture the complexity of domain shifts. To address this, our approach integrates a domain estimator that learns to assign experts based on the data itself, allowing the model not only to bypass reliance on domain labels at inference time but also to surpass the performance of models using predefined domain labels. This solution addresses a key limitation highlighted in (Zhong et al., 2022), increasing the model's flexibility and practicality in real-world applications.

Our contributions are summarized as follows: (1) We introduce a unified framework to address the domain shifts challenge in both centralized and federated learning, based on mixtures of multi-dimensionally indexed domains. (2) We propose a novel approach, EMoE, to effectively manage multi-dimensional domain shifts, requiring only partial or no domain labels during training and testing. This method is naturally extended to federated learning, ensuring robust generalization across both training and unseen testing sites. (3) We integrate a domain estimator that assigns experts based on the data itself, enabling our model to overcome the limitations of missing or unreliable domain labels, further improving its adaptability. (4) We demonstrate the effectiveness of our method across five centralized learning tasks and one real-world federated learning scenario, outperforming existing DG, PFL, and FDG methods. Our approach comprehensively handles various domain shifts and generalizes effectively to unseen data distributions without the need for domain labels.

## 2 RELATED WORK

### 2.1 DOMAIN GENERALIZATION

In domain generalization, training data from one or more source domains is used while test data originates from different, unseen target domains. This approach, distinct from domain adaptation, does not involve target domain data during training, as outlined by Quinonero-Candela et al. (2008). We explore scenarios where training data consists of diverse, multi-dimensional domains, acknowl-

edging that domain shifts are multifactorial and these domains are identified and known to some extent. Koh et al. (2021) differentiates "domain generalization" from "subpopulation shift," where training and test domains overlap but vary in composition. Our study extends the concept of DG to include both scenarios, in line with Gulrajani & Lopez-Paz (2020); Shi et al. (2021).

Approaches in DG include: i) Domain-invariant representation learning, which seeks to extract features making domains indistinguishable for prediction, inspired by generalization error bounds (Ganin et al., 2016; Ben-David et al., 2010). Techniques include domain prediction penalization (Ganin et al., 2016; Wang et al., 2019a; Huang et al., 2020), feature distribution matching across domains (Sun & Saenko, 2016; Li et al., 2018a), domain gradient alignment (Koyama & Yamaguchi, 2020; Shi et al., 2021), data augmentation (Yue et al., 2019; Xu et al., 2020; Yao et al., 2022b), and learning through pretext tasks (Carlucci et al., 2019). ii) Invariant Risk Minimization (IRM) by Arjovsky et al. (2020), aims for a universal representation where domain-specific optimal classifiers are identical, addressing invariant causal relationships and domain-specific noise, with extensions like IB-IRM (Ahuja et al., 2021). iii) Distributional Robustness, which minimizes worst-case losses across data distributions from different training domains, addressing covariate shifts and subpopulation shifts, discussed in Rojas-Carulla et al. (2015); Quiñonero-Candela et al. (2008); Hu et al. (2018); Sagawa et al. (2019).

## 2.2 PERSONALIZED FEDERATED LEARNING

Personalized federated learning has been explored through various approaches, each offering unique strategies for tailoring models to individual clients. These approaches include: i) Clustered FL, which divides clients into clusters to develop an optimal model for each group, as discussed in Sattler et al. (2020); Mansour et al. (2020); Ghosh et al. (2020). ii) Meta-learning techniques, which adapt the model to new clients using prior knowledge, covered in works like Chen et al. (2018); Fallah et al. (2020); Jiang et al. (2019); Khodak et al. (2019). iii) Local and global model interpolation, an approach that combines local client models with a global model, as seen in Deng et al. (2020); Corinzia et al. (2019); Mansour et al. (2020). iv) Multi-Task Learning (MTL), where separate but related tasks are learned simultaneously, as explained in Vanhaesebrouck et al. (2017); Smith et al. (2017); Zantedeschi et al. (2020) and further explored in Hanzely & Richtárik (2020); Hanzely et al. (2020); T Dinh et al. (2020); Huang et al. (2021); Li et al. (2021a). v) Local fine-tuning methods, where individual models are adjusted post-training, as presented in Wang et al. (2019b); Yu et al. (2020). vi) Developing local representations or heads for individual clients, a strategy outlined in Arivazhagan et al. (2019); Liang et al. (2020); Collins et al. (2021a). vii) Creating personalized models using hypernetworks or supermodels, as proposed in Shamsian et al. (2021); Chen & Chao (2021); Xu et al. (2022).

## 2.3 FEDERATED DOMAIN GENERALIZATION

In contrast to PFL, Federated Domain Generalization (FDG) aims to enhance model performance on unseen clients. Research in this area is limited; Liu et al. (2021) employ amplitude spectrum analysis for data distribution as information to exchange, increasing costs and privacy risks. Jiang et al. (2022) introduce a flatness-aware optimization, while Zhang et al. (2023) adjust aggregation weights based on domain generalization gaps. Xu et al. (2022) combine PFL techniques by selecting the most similar personalized local model for an unseen client.

Most existing PFL and FDG approaches treat each client as a distinct domain. However, this often overlooks the significant interconnections in data distributions across different sites. Recent studies, such as those by Marfoq et al. (2021) and Wu et al. (2023), propose that the data distribution at each client's end is more accurately represented as a mixture of several underlying unknown distributions. Nevertheless, these studies do not fully exploit the available domain information.

Our work extends the data distribution assumptions from Zhong et al. (2022) in FL and Koh et al. (2021) in centralized settings, treating data as mixtures of predefined domains. By considering the multifactorial domain shifts, we concentrate on scenarios where the data distributions are characterized as mixtures of multi-dimensional indexed domains. Our model's design is crafted to be readily applicable to any new data distribution. This approach allows for a more nuanced understanding of domain shifts, offering avenues for developing models that are both more robust and adaptable to diverse data landscapes.

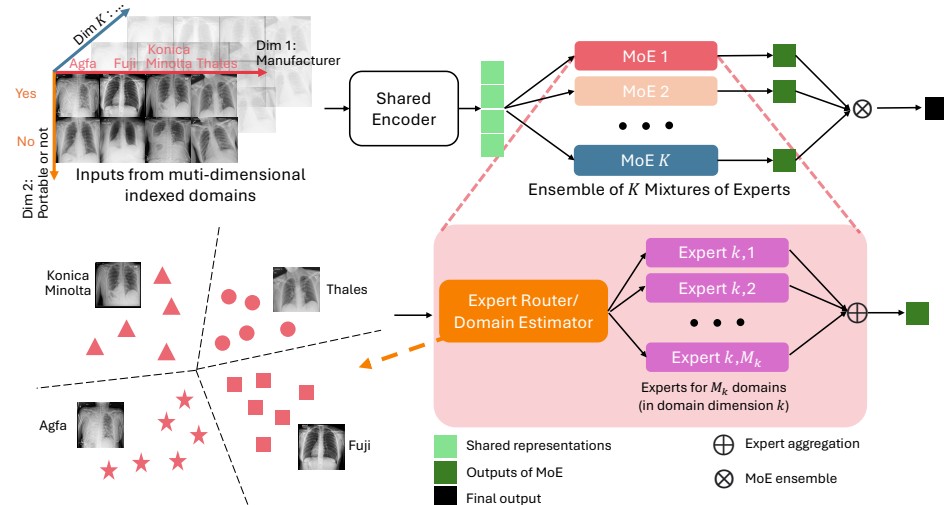

Figure 2: Illustration of our EMoE framework. Each sample is assigned to an expert via the domain estimator and each MoE addresses domain shifts within each specific domain dimension. These MoEs are then ensembled across various domain dimensions, producing the final model outputs.

## 3 PROBLEM AND METHODOLOGY

In this section, we begin by formalizing our problem setting, followed by the introduction our proposed solution, EMoE. We break down the framework of EMoE into four parts: (1) the learning objective, (2) the MoE ensemble design, (3) the domain estimator, and (4) the learning algorithms.

### 3.1 PRELIMINARY

We address a setting where one predicts the label $y \in \mathcal{Y}$ from the input feature $x \in \mathcal{X}$. We assume the data are generated from multi-dimensional domains. Specifically, there are $K$ dimensions of domains with $\{M_1, M_2, \cdots, M_K\}$ predefined domains in each dimension. For example, gender can be one dimension with male and female as two domains, and race can be another dimension with white, black, Asian as three domains. We denote the domain label by a domain vector $z = [z_1, \cdots, z_K] \in \mathcal{Z} = [M_1] \times \cdots \times [M_K]$ that indicates which domains this sample belongs to (for each domain dimension). Our goal is to learn a predictor that works for any domains at any domain dimension. Thus, we care not only about the average performance but the worst domain performance as well. Following Zhong et al. (2022); Koh et al. (2021), we regard each unique domain vector $z$ to be associated with a data distribution $\mathcal{D}_z$. Then we can formulate any data distribution as the mixture of multi-dimensional domains, i.e., $\mathcal{D} = \sum_{z \in \mathcal{Z}} P(z)\mathcal{D}_z$, where $\{P(z)\}$ denotes the mixture coefficients.

In **Centralized Learning**, domain generalization aims to train a model on a training distribution $\mathcal{D}_{tr} = \sum_{z \in \mathcal{Z}} P_{tr}(z)\mathcal{D}_z$. The goal is for this model to generalize effectively to a distinct test distribution $\mathcal{D}_{ts} = \sum_{z \in \mathcal{Z}} P_{ts}(z)\mathcal{D}_z$, where the training and test domain mixture coefficients, $\{P_{tr}(z)\}$ and $\{P_{ts}(z)\}$, are not identical. In **Federated Learning**, the aim is to develop a model that learns from a diverse set of client-specific data distributions, formulated as $\mathcal{D}_i = \sum_{z \in \mathcal{Z}} P_i(z)\mathcal{D}_z$ for each client. With $n$ representing the total number of clients, where each client is indexed by $i \in [n] \equiv \{1, 2, ..., n\}$, the challenge lies in enabling this model to generalize effectively not just across the participating clients during training but also to any unseen clients characterized by $\mathcal{D}_{uk} = \sum_{z \in \mathcal{Z}} P_{uk}(z)\mathcal{D}_z$, where $\{P_{uk}(z)\}$ remains unknown.

### 3.2 LEARNING OBJECTIVE WITH DOMAIN REWEIGHTING

Given a task, the task-specific loss is defined by $\mathcal{L}_{\texttt{task}} : \mathcal{Y} \times \mathcal{Y} \to \mathbb{R}$. A model, represented as $f : \mathcal{X} \times \mathcal{Z} \to \mathcal{Y}$, makes predictions based on inputs and domain labels. Our method focuses on achieving equitable minimization of risks, formulated as $\mathcal{R}_z(f) := \mathbb{E}_{(x,y) \sim \mathcal{D}_z} \mathcal{L}_{\texttt{task}}(f(x, z), y)$,

across each multi-dimensionally indexed domain $\boldsymbol{z}$. This is key to achieving strong generalization, even when the mixture coefficients $\{P(\boldsymbol{z})\}$ differ significantly from those in the training set.

In centralized learning, this goal can be straightforwardly accomplished through reweighting samples in different domains. However, in federated learning, this straightforward approach is not feasible due to the distribution of data across various clients. Therefore, we introduce proper weights for each training sample at different local clients. Specifically, suppose each client $i$ has a collection of data samples $\mathcal{S}_i = \{(\boldsymbol{x}^j, y^j, \boldsymbol{z}^j)\}_{j=1}^{L_i}$, we use $\mathcal{S}_{i,k,m}$ to denote the set of samples from domain $m$ at dimension $k$ in client $i$, with $L_{i,k,m} := |\mathcal{S}_{i,k,m}|$ denoting the sample size. Further, $L_i := \sum_{m=1}^{M_k} L_{i,k,m}, \forall k$ is the number of samples in client $i$ while $L_{k,m} := \sum_{i=1}^{n} L_{i,k,m}$ is the total number of samples belonging to domain $m$ at dimension $k$ across all the clients. We denote the empirical risk at client $i$ specific to domain $m$ at dimension $k$ as $\hat{\mathcal{R}}_{i,k,m}(f)$. Following a similar approach as in Zhong et al. (2022), by using $u_{k,m} = \frac{L}{L_{k,m} \cdot M_k}$ to reweight the risk for each domain at each dimension, we derive our learning objective:

$$\hat{\mathcal{R}}(f) := \sum_{i=1}^{n} \frac{L_i}{L} \hat{\mathcal{R}}_i(f) = \frac{1}{K} \sum_{k=1}^{K} \frac{1}{M_k} \sum_{m=1}^{M_k} \hat{\mathcal{R}}_{k,m}(f), \tag{1}$$

where $\hat{\mathcal{R}}_{k,m}(f) := \sum_{i=1}^{n} \frac{L_{i,k,m}}{L_{k,m}} \hat{\mathcal{R}}_{i,k,m}(f)$. Our objective treats each domain within every dimension equally, whereas standard FL treats each client equally, ignoring the latent domain structure. In practice, our objective can be implemented simply by weighting samples individually using $u = \frac{1}{K} \sum_{k=1}^{K} \frac{L}{L_{k,m} \cdot M_k}$ when computing the loss.

### 3.3 Ensemble of Mixtures of Experts

The design of EMoE is motivated by the need to effectively handle the complexity of multi-dimensional domain shifts, which are common in real-world scenarios. Fig. 2 illustrates the overview of our EMoE framework, consisting of a shared encoder (representation extractor) $\phi$ and $K$ distinct Mixtures of Experts (MoEs). Each MoE $k$ is specialized to address domain dimension $k$ by incorporating a mixture of $M_k$ experts, labeled as $h_{k,m}$ for each $m \in [M_k]$. This modular design enables each MoE to focus on specific factors contributing to domain shifts, allowing the model to adapt flexibly to various domain combinations. By isolating and addressing shifts across multiple dimensions, EMoE enhances the model's ability to generalize effectively to unseen domains. Additionally, this architecture is versatile and can be easily integrated with a wide range of existing models, making it both effective and practical for addressing complex domain shifts.

Specifically, it first computes the output through corresponding MoE for each domain dimension, i.e., $o_k := \sum_{m=1}^{M_k} \tilde{z}_{k,m} o_{k,m}$, where $o_{k,m} := h_{k,m}(\phi(x))$ is output of expert $k, m$ and $\tilde{z}_{k,m}$ is the expert assignment score used to choose proper experts. If the true domain label is accessible, it can be directly used as expert assignment score with one hot embedding, $\tilde{z}_k = \texttt{one\_hot}(\boldsymbol{z}_k)$. Otherwise, we adopt an domain estimator/expert router (orange box in Fig. 2) to estimate an expert assignment scores $\hat{z}_k$.

To compile the final model output, we apply an aggregation function $\texttt{AGG}(\cdot)$ to ensemble the outputs from all MoEs, i.e., $o = \texttt{AGG}(\{o_k\}_{k=1}^{K})$. We define three aggregation functions: $\texttt{AVG}$, $\texttt{MAX}$ and $\texttt{PRO}$. The $\texttt{AVG}$ function refers to averaging the MoE's outputs as the final prediction. The $\texttt{MAX}$ function refers to using the MoE with the highest confidence level (largest prediction probability for multi-class classification) as the final output.

$$\texttt{AVG}: \; o = \frac{1}{K} \sum_{k=1}^{K} o_k \qquad \texttt{MAX}: \; o = o_{k^*}, \text{ where } k^* = \arg\max_k \texttt{conf}(o_k) \tag{2}$$

**Probabilistic Aggregation** $\texttt{PRO}$ is a prediction aggregation method under the assumption that different domains have *independent* effect to the relation between $x$ and $y$. Specifically, we assume $z_1, \cdots, z_K$ are mutually independent given either $x$ or $x, y$, i.e., $p(z_1, \cdots, z_K|x) = \prod_{k=1}^{K} p(z_k|x)$ and $p(z_1, \cdots, z_K|x, y) = \prod_{k=1}^{K} p(z_k|x, y)$. Under the assumption, we can show that,

$$p(y|x, z_1, \cdots, z_K) = \prod_{k=1}^{K} p(y|x, z_k) \bigg/ p(y|x)^{K-1} \tag{3}$$

Inspired by the above equation, we design a new aggregation

$$\text{PRO:} \quad o = \sum_{k=1}^{K} o_k - (K-1)o_{\text{base}}, \tag{4}$$

where $o_{\text{base}}$ is the output of an unconditioned predictor which does not consider domains.

### 3.4 DOMAIN ESTIMATOR/EXPERT ROUTER

In practical scenarios, it is common for specific domain labels for some data samples to be missing, unreliable, or unable to accurately characterize domain information. For example, a patient may choose not to disclose their demographics, or the available domain labels might not fully capture the complexities of the data. This issue often arises during both the training and testing phases. Missing or inaccurate domain information may result from initial data collection or from data originating in a new, previously unseen domain, where no corresponding domain expert exists. To address these challenges, our framework incorporates a domain estimator to autonomously identify the domain of the data, enabling the allocation of a suitable expert for prediction.

**During training**, the domain estimator calculates a set of domain assignment scores $\{\hat{z}_k\}$ to assign each sample to an expert via `Gumbel-Softmax` (Jang et al., 2016; Maddison et al., 2016):

$$\bar{z}_{k,m} = \frac{\exp(l_{k,m} + \gamma_m)}{\sum_{j=1}^{M_k} \exp(l_{k,j} + \gamma_j)}, \tag{5}$$

where $l_{k,j}$ is the logits computed by the estimator for $j$-th domain of $k$-th dimension, and $\{\gamma_j\}$ are i.i.d random samples drawn from the `Gumbel`$(0,1)$ distribution. We assign the expert using one-hot operation with argmax over all the domains. Since one-hot operation with argmax is not differentiable, we use the straight through trick in (Van Den Oord et al., 2017) to calculate the assignment score:

$$\hat{z}_k = \texttt{one\_hot}(\bar{z}_{k,\text{argmax}}) + \bar{z}_k - \texttt{sg}(\bar{z}_k), \tag{6}$$

where `sg` is the stop gradient operator. We denote this one-hot assignment strategy as *hard assignment*. An alternative way is to directly set $\hat{z}_k = \bar{z}_k$, which we referred as *soft assignment*. Empirically we find that hard assignment outperforms soft assignment (as shown in Table 3 and Table 12). **During inference**, the scores are calculated through plain `soft_max`.

The domain estimator can be effectively trained using direct supervision with the cross-entropy loss, denoted as $\mathcal{L}_{\text{domain}} = L_{\text{ce}}(l_k, \tilde{z}_k)$, when domain labels are available. Alternatively, it can also be trained using $\mathcal{L}_{\text{task}}$ on the model's final output, as the domain estimator is a differentiable component of the overall model. Empirically, we found that initially training the model with both $\mathcal{L}_{\text{domain}}$ and $\mathcal{L}_{\text{task}}$ for each MoE, followed by training with only $\mathcal{L}_{\text{task}}$ on the final output, yields the best results. We refer to this first phase as the pretraining of EMoE. This approach is advantageous because it leverages domain-related information directly from the data itself, rather than relying solely on predefined domain labels, which may not fully capture the underlying domain characteristics. More discussion can be found in Section 4.2.

### 3.5 LEARNING ALGORITHMS

**Centralized Learning.** The training process consists of two phases: pretraining and training, summarized in Algorithm 1. In the pretraining phase, the model is updated using two losses: $\mathcal{L}_{\text{domain}}$ for the domain estimator output $l_k$ and $\mathcal{L}_{\text{task}}$ for the MoE output $o_k$, aiming to establish a strong initialization. This phase uses possibly unreliable predefined domain labels. In the subsequent training phase, the model focuses solely on $\mathcal{L}_{\text{task}}$ applied to the final output $o$, discarding the predefined domain labels and training all components end-to-end based on data-driven domain estimation.

**Federated Learning.** We decouple the learning of the shared encoder and MoEs to stabilize the learning procedure. At each communication round, we first update the domain experts locally using the specific task loss $\mathcal{L}_{\text{task}}$, and then aggregate them at the server. Next, we update the shared representation and domain estimators by minimizing $\mathcal{L} = \mathcal{L}_{\text{task}} + \lambda \mathcal{L}_{\text{domain}}$, where $\lambda$ is a hyperparameter. The learning procedure is summarized in Algorithm 2.

---

**Algorithm 1** EMoE

---

**Input:** Data set $\mathcal{S}$; pretraining epochs $T_{pre}$; total epochs $T$.
**for** $t = 1$ **to** $T_{pre}$ **do**
    Pretrain model using domain estimation loss $\mathcal{L}_{\texttt{domain}}$ based on domain estimator outputs $\{l_k\}$
    and task-specific loss $\mathcal{L}_{\texttt{task}}$ on the outputs of MoEs $\{o_k\}$.
**end for**
**for** $t = T_{pre} + 1$ **to** $T$ **do**
    Train model using task-specific loss $\mathcal{L}_{\texttt{task}}$ based on the final aggregated output $o$.
**end for**

---

**Algorithm 2** FEDEMoE

---

**Input:** Data $\mathcal{S}_{1:n}$; number of local updates $\tau_h$ for the experts, $\tau_\phi$ for representation; number of communication rounds $T$; learning rate $\eta$.
Initialize representation, estimators and experts $\phi^0, \{e_k^0\}, \{h_{k,m}^0\}$.
**for** $t = 1, 2, ..., T$ **do**
    Server sends $\phi^{t-1}, \{e_k^{t-1}\}, \{h_{k,m}^{t-1}\}$ to the $n$ clients;
    **for** client $i = 1, 2, ..., n$ **in parallel do**
        update experts $\{h_{k,m}^t(i)\}$ with $\tau_h$ epochs of LocalTraining($\phi^{t-1}, \{e_k^{t-1}\}, \{h_{k,m}^{t-1}\}$) using
        $\mathcal{L}_{\texttt{task}}$
        Client $i$ sends updated $\{h_{k,m}^t(i)\}$ to the server.
    **end for**
    Server aggregate the experts, update $\{h_{k,m}^t\}$ and send them back to $n$ clients;
    **for** client $i = 1, 2, ..., n$ **in parallel do**
        update domain estimator $\{e_k^t(i)\}$, shared representation $\{\phi^t(i)\}$ with $\tau_\phi$ epochs of
        LocalTraining($\phi^{t-1}, \{e_k^t\}, \{h_{k,m}^t\}$) using $\mathcal{L}$
        Client $i$ sends updated $\{e_k^t(i)\}, \{\phi^t(i)\}$ to server.
    **end for**
    Server updates representation and estimators $\phi^t \leftarrow \sum_{i=1}^n \frac{L_i}{L} \times \phi^t(i), e^t \leftarrow \sum_{i=1}^n \frac{L_i}{L} \times e^t(i)$.
**end for**

---

Table 1: Experiment setups. All tasks are binary classification.

| Dataset | Domains | Classification Task |
|---------|---------|---------------------|
| CMNIST | 2 digit colors x 2 background colors | digit (0,1,2,3,4) vs. digit (5,6,7,8,9) |
| CelebA | 2 genders x 2 age groups | blonde vs. non-blonde |
| FairFace | 2 genders x 7 races | age$<$40 vs. age$\geq$40 |
| EXAM | portability x 6 manufacturers x 5 races | severe symptoms vs. not |

## 4 EXPERIMENTS

Our experimental evaluation consists of two parts. First, we assess our methodology in a centralized learning setting, using three binary classification datasets characterized by two-dimensional domain shifts. This part is aimed at showcasing the enhanced capability of our method in handling multi-dimensional domain shifts, compared to existing methods that tackle simpler domain shifts scenarios. Our approach is adaptable to a range of tasks, due to space constraints, results for image regression and segmentation in the centralized setting are provided in Appendix B.

Next, we focus on a real-world FL dataset. This section aims to demonstrate the superiority of our method over previous approaches designed for heterogeneous data distributions in FL settings. The employed dataset offers a robust platform to highlight the effectiveness of our model in a more complex and realistic FL environment.

### 4.1 EVALUATION IN CENTRALIZED LEARNING

**Datasets.** In our centralized learning evaluation, we utilize three datasets: Colored MNIST (CMNIST), CelebA Liu et al. (2015), and FairFace Karkkainen & Joo (2021). Domains and label information are presented in Table 1. CMNIST is crafted to feature digit color and background color as its two-dimensional domains, with the binary label determined by the digit shifting depending on

Table 2: Performance on centralized learning with multi-dimensional domain shift. Each cell reports the worst group accuracy (%). We consider groups defined only by the first domain dimension (dim1) or the second dimension (dim2) as well as two dimension jointly (all).

| | | CMNIST | | | CelebA | | | FairFace | | |
|---|---|---|---|---|---|---|---|---|---|---|
| | | dim1 | dim2 | all | dim1 | dim2 | all | dim1 | dim2 | all |
| ERM | | 97.4 | 97.6 | 96.0 | 79.1 | 90.3 | 25.8 | 80.4 | 80.3 | 45.0 |
| UW | | 97.4 | 97.8 | 96.8 | 90.2 | 92.5 | 73.1 | 83.4 | 83.6 | 72.0 |
| RIDG | | 98.2 | 98.4 | 97.1 | 90.2 | 92.6 | 71.5 | 84.1 | 84.6 | 73.1 |
| SAGM | | 98.3 | 98.3 | 97.3 | 90.8 | 92.8 | 80.1 | 84.5 | 84.3 | 74.1 |
| IRM | dim1 | 97.6 | 97.9 | 96.3 | 90.7 | 92.6 | 72.6 | 83.6 | 83.8 | 72.1 |
| | dim2 | 97.9 | 97.9 | 96.6 | 88.6 | 93.1 | 63.4 | 83.2 | 83.7 | 66.4 |
| | all | 97.8 | 97.9 | 97.0 | 90.3 | 92.5 | 72.6 | 83.0 | 83.1 | 70.2 |
| LISA | dim1 | 97.5 | 97.8 | 96.4 | 91.1 | 92.2 | 80.0 | 82.9 | 82.7 | 55.8 |
| | dim2 | 97.6 | 97.5 | 96.2 | 86.4 | 92.5 | 50.5 | 82.4 | 83.1 | 61.6 |
| | all | 98.0 | 97.9 | 96.9 | 90.8 | 92.2 | 77.4 | 82.5 | 82.6 | 71.2 |
| GroupDRO | dim1 | 97.4 | 97.9 | 96.6 | 90.7 | 92.3 | 73.1 | 83.3 | 83.0 | 73.0 |
| | dim2 | 98.0 | 97.9 | 96.6 | 87.8 | 93.0 | 59.2 | 83.2 | 83.6 | 67.4 |
| | all | 97.9 | 98.0 | 96.5 | 89.7 | 91.8 | 76.9 | 82.5 | 82.9 | 73.7 |
| **EMoE (ours)** | avg | 98.5 | 98.3 | 97.4 | 91.5 | 92.2 | 81.1 | 82.7 | 83.2 | 71.8 |
| | max | 98.2 | 98.3 | **97.5** | 91.4 | 92.2 | 82.3 | 83.1 | 83.7 | 72.1 |
| | pro | 98.4 | 98.3 | 97.1 | 92.1 | 92.5 | **83.9** | 83.9 | 83.8 | **75.1** |

both domain dimensions. CelebA and FairFace, both face recognition datasets, use various attributes to define the dimensions of their respective domains, with domain shifts inherent to the data. The comprehensive details of domain shifts for each dataset are available in Appendix A.

**Evaluation Protocols.** In line with the approach in Sagawa et al. (2019), we employ worst-group accuracy to assess the performance of all methods. To ensure a thorough evaluation relative to different domain dimensions, we define groups variably. For domain dimension $k$, we define groups as $g = (d_k, y)$ where $d_k$ is the domain index at dimension $k$ to evaluate performance specific to that dimension. We also use $g = (d_1, \ldots, d_K, y)$ for an evaluation that considers all dimensions collectively. The hyper-parameters and training details are listed in Appendix A.

**Baselines.** We compare our approach against Empirical Risk Minimization (ERM) and several domain generalization methods: IRM (Arjovsky et al., 2020), GroupDRO (Sagawa et al., 2019), LISA (Yao et al., 2022b), RIDG (Chen et al., 2023), and SAGM (Wang et al., 2023), in addition to upweighting (UW), which is widely recognized for addressing subpopulation shifts. To ensure a fair comparison, we apply upweighting to all methods except ERM, as the imbalance in group distributions significantly affects performance. For algorithms such as IRM, GroupDRO, and LISA, which leverage domain labels, we perform evaluations using different domain definitions $\hat{d}$. This includes $\hat{d} = d_k$ for each $k$ in $|K|$, focusing on single-dimensional domain information, as well as $\hat{d} = (d_1, \ldots, d_K)$, which considers all dimensions collectively but treats them as separate categories, disregarding their inherent multi-dimensional structure.

**Results.** Table 2 compares the performance of EMoE method against established baselines across the CMNIST, CelebA, and FairFace datasets. The baseline models, which incorporate domain information, generally show enhanced performance on the specific domain dimensions they were tailored for. However, utilizing all domain dimensions without domain structure awareness sometimes can hurt the performance. For example for LISA, using ALL dimensional domains is worse than using only the first dimension of domain in CelebA dataset. In contrast, EMoE method demonstrates superior performance consistently across all three datasets and for different domain evaluations. This underscores the robustness of EMoE in handling multi-dimensional domain shifts. More results, including experiments on additional datasets and detailed ablation studies, are in Appendix B.

**Effects of Aggregation Methods.** The PRO aggregation method significantly outperforms other methods in the CelebA and FairFace datasets, confirming the hypothesis that domain dimensions operate independently. Specifically, in CelebA, the attributes of age (young/old) and gender manifest as independent variables when considered against hair color. Similarly, in FairFace, the attributes of race and gender prove to be independent when conditioned on age. These findings validate the

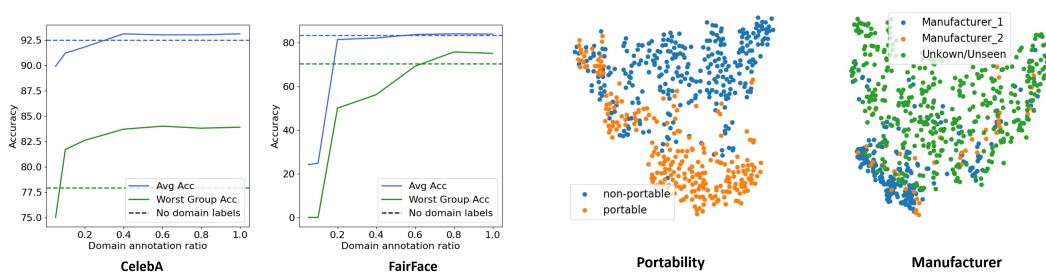

(a) Results with partial domain annotation          (b) The t-SNE visualization of domains.

Figure 3: (a)Effectiveness of our method with limited domain label annotations, demonstrated on CelebA and FairFace datasets. (b) The t-SNE visualization of the input of the domain estimators on the three independent test sites.

Table 3: AUC results on independent test sites.

| Method | Site1 | Site2 | Site3 | Avg |
|---|---|---|---|---|
| FedAvg McMahan et al. (2017) | $.802\pm.020$ | $.892\pm.006$ | $.835\pm.009$ | $.843\pm.008$ |
| FedProx Li et al. (2018b) | $.813\pm.010$ | $.891\pm.002$ | $.880\pm.019$ | $.861\pm.007$ |
| FedDG_GA Zhang et al. (2023) | $.818\pm.009$ | $.881\pm.004$ | $.832\pm.026$ | $.844\pm.011$ |
| FedDAR Zhong et al. (2022) | $.824\pm.009$ | $.910\pm.003$ | $.890\pm.007$ | $.874\pm.004$ |
| FedEMoE (Single) | $.850\pm.003$ | $.917\pm.001$ | $.912\pm.005$ | $.893\pm.002$ |
| FedEMoE (Soft) | $.844\pm.004$ | $.917\pm.003$ | $\mathbf{.924\pm.009}$ | $.895\pm.003$ |
| FedEMoE (Hard) | $\mathbf{.854\pm.004}$ | $\mathbf{.921\pm.001}$ | $.920\pm.005$ | $\mathbf{.899\pm.002}$ |
| FedEMoE (GT) | $.840\pm.004$ | $.917\pm.003$ | $.877\pm.012$ | $.878\pm.004$ |

assumption underpinning our domain dimensionality strategy and reinforce the efficacy of PRO aggregation in our EMoE framework.

**Effects of Missing Domain Labels.** We assess our model's capacity to handle missing domain labels under varying levels of annotation availability, as depicted in Figure 3a. The dashed line illustrates the performance of an identical model architecture trained without any domain labels. Our method continues to operate effectively despite the scarcity of domain labels. However, at lower label ratios, there is a noticeable decline in performance, attributable primarily to insufficient pretraining of the domain estimator, especially within underrepresented domains.

## 4.2 Evaluation in Federated Learning

**Dataset.** We conduct experiments on the EXAM dataset (Dayan et al., 2021), which is a large-scale, real-world healthcare FL study. Our training/validation dataset consists of a portion of the dataset, which includes 6 sites and a total of 7,681 cases, while our test dataset includes 3 independent sites and 1,454 cases. This dataset contains electronic medical records (EMR) and Chest X-rays (CXR) of patients suspected of COVID-19 in emergency departments (ED). The task is to predict the need for oxygen therapy exceeding high-flow oxygen within 72 hours, signifying severe symptoms. We use the same data preprocessing and model architecture as in the original work (Dayan et al., 2021).

**Model Specifications & Evaluation Protocols.** Our model for the EXAM dataset includes three MoEs, reflecting the $K = 3$ domain dimensions: (1) CXR device portability, (2) manufacturer, and (3) patients' race group, with $M_1 = 2$, $M_2 = 6$, and $M_3 = 5$. It is important to note that in practice, some domains may be unknown or have too few samples; in such cases, we group them as a single domain and assign an additional expert. All experiments were run with 5 random seeds, reporting the mean and standard error of the AUC, averaged over the last five communication rounds, following prior work (Collins et al., 2021b; Zhong et al., 2022).

**Federated Domain Generalization on Unseen Test Sites.** We first evaluate our method on three independent test sites that are not seen during model training, comparing it with FedAvg, FedProx, FedDAR, and FedDG_GA. To ensure fairness, no domain labels are provided to the models. Note that in this experiment, we do not include client-wise PFL methods as baselines, as there are no standard protocols to apply them to external test sites. These methods can personalize the model

Table 4: AUCs results on different domain dimensions with 5-fold cross validation.

| Method | Portable (P) | | Manufacturer (M) | | Race group (R) | | Client Avg |
|---|---|---|---|---|---|---|---|
| | Worst | Avg | Worst | Avg | Worst | Avg | |
| FedAvg | .860±.009 | .882±.005 | .816±.017 | .886±.007 | .843±.013 | .898±.010 | .886±.006 |
| FedProx | .860±.006 | .876±.005 | .812±.009 | .879±.006 | .829±.014 | .896±.010 | .880±.005 |
| FedDG_GA | .846±.006 | .862±.007 | .802±.009 | .869±.008 | .825±.009 | .881±.007 | .860±.008 |
| FedBN Li et al. (2021b) | .798±.006 | .831±.004 | .802±.012 | .860±.004 | .732±.040 | .830±.015 | .868±.003 |
| FedPer Arivazhagan et al. (2019) | .857±.011 | .878±.009 | .814±.016 | .880±.010 | .842±.015 | .893±.011 | .879±.009 |
| FedRep Collins et al. (2021a) | .864±.006 | .872±.008 | .816±.011 | .879±.006 | .840±.012 | .881±.011 | .880±.006 |
| LG-FedAvg Liang et al. (2019) | .830±.005 | .877±.007 | .811±.016 | .858±.009 | .814±.024 | .880±.014 | .867±.008 |
| FedDAR | .888±.005 | .900±.004 | .863±.006 | .902±.004 | **.856±.008** | **.915±.006** | .901±.003 |
| **FedEMoE (Ours)** | **.895±.005** | **.906±.003** | **.874±.006** | **.914±.004** | **.863±.009** | **.918±.004** | **.909±.003** |

for each client involved in the training, but cannot be trivially applied to a new client. The results presented in Table 3, show that our model generalizes well to unseen sites under FL, even without any domain information. FedEMoE consistently achieves 2% to 2.5% higher AUC than all baselines.

**Effectiveness of Domain Estimator.** We evaluate different variants of our model to assess the effectiveness of various strategies for handling missing domain labels, as shown in Table 3: (1) using only the expert assigned to the unknown domain (**single**), (2) applying soft assignment (**soft**), and (3) applying hard assignment (**hard**). For reference, we also include the results of EMoE when using ground truth domain labels (**GT**). Our experiments reveal that the *hard assignment* strategy consistently outperforms the others. Notably, the model demonstrated superior performance with estimated domain assignments over ground truth domain labels, particularly for Site3. We find that domain labels alone do not fully capture the data distribution, as they may overlook critical factors, especially in medical data, where a patient's condition cannot easily be defined by a few labels. Although most samples in Site3 come from a manufacturer seen during training, the images appear significantly different from those at the training sites due to varying imaging protocols or device models. *Domain shifts exists even within the same data group.* In our method, domain labels initially serve as a preliminary cluster assignment during training. As training progresses, the model accurately assigns each sample to the appropriate expert, aligning more closely with the actual data distributions. This approach has consistently improved performance across multiple experiments, as confirmed by the ablation study of pretraining in a centralized setting (Appendix B.3). Abandoning the ground truth domain labels after a period of domain estimator pretraining yields the best results.

Figure 3 employs t-SNE (Van der Maaten & Hinton, 2008) to depict the domain estimators' learned input features. The figure illustrates that while the domain dimension *portable* is more distinct, overlaps remain. In the domain dimension of manufacturers, the unknown/unseen domain distribution appears significantly dispersed. Compared to *manufacturer_1*, which exhibits a tighter clustering, *manufacturer_2* shows a more widely spread distribution. These results further confirm the superiority of estimating the domain from the data itself over relying solely on the given domain labels.

**Effectiveness of MoEs Ensemble** The effectiveness of the MoEs ensemble is evaluated through 5-fold cross-validation across six training sites. The unseen test sites are excluded from domain-wise evaluation due to an inadequate number of samples. We compare our method, FedEMoE , with FL baselines FedAvg, FedProx, client-wise PFL methods FedBN, FedRep, FedPer, and LG-FedAvg, domain-wise PFL methods FedDAR, and a federated DG approach FedDG_GA. Results summarized in Table 4 reveal that FedEMoE consistently surpasses the baseline models in terms of both average and minimum AUC scores, across all evaluated domains and clients. Compared with the results of FedDAR which only focus on race domains, the ensemble of multiple MoEs notably improves performance along two other domain dimensions, which elevates the overall generalizability.

## 5 CONCLUSION

In this study, we introduced EMoE, a domain generalization method to address multi-dimensional domain shifts in both centralized and federated learning. Unlike prior methods focusing on single dimensions or isolated domains, EMoE handles shifts across multiple dimensions, ensuring strong generalization. Our approach uses an Ensemble of Mixtures of Experts (EMoE), with each MoE specializing in a domain dimension. We also integrated a domain estimator to manage missing or unreliable labels, enhancing flexibility and practicality. Extensive experiments across six datasets demonstrated EMoE's superior performance, outperforming state-of-the-art DG, PFL, and FDG methods. Particularly in federated learning, FedEMoE significantly improves fairness and robustness, setting a new benchmark for federated domain generalization research.

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

## A  DATASETS AND IMPLEMENTATION DETAILS

### A.1  DATASET DETAILS

**Colored MNIST (CMNIST)**   We use the digit and background colors as two domain dimensions. The task involves classifying digits into 2 classes, with conditional shifts across domains. In the default setting, where digits are red and backgrounds are black, the classes are (0,1,2,3,4) vs (5,6,7,8,9). When the digit color changes to green, the labels for digits 2 and 5 will flip. With a white background, the labels for digits 4 and 9 will flip. In the training set, the proportion of red to green samples is set at 8:2, and the proportion of black to white backgrounds is also 8: 2. For the validation set, the proportions are equal to 1:1 for all domain dimensions. In the test set, the proportion of red to green samples is 1:9, and the proportion of black to white backgrounds is also 1:9. The data sizes of train, validation, and test sets are 30000, 10000, and 20000, respectively.

**CelebA**   Adhering to the preprocessing steps outlined in  Sagawa et al. (2019), we utilize CelebA's celebrity face images. The objective is to categorize hair color as either "blond" or "not blond". Gender and age are employed as two domain dimensions. Notably, the dataset exhibits a significant imbalance in group distribution. Within the training set, the most populous group is (not blond, female, young) with 64,036 samples, whereas the smallest group is (blond, male, old), comprising merely 463 samples.

**FairFace (Kärkkäinen & Joo, 2019)**   FairFace is a publicly available face attribute dataset that ensures a balanced representation of race, gender, and age for bias analysis. It features seven racial groups, nine age groups, and two genders. In this dataset, gender and race serve as the domain dimensions. The classification task involves determining whether an individual's age is 40 or older. The original dataset comprises 66,744 samples in the training set and 10,954 samples in the validation set. We evenly partition the initial validation set, using the first 50% for validation purposes and the remaining 50% for testing.

**EXAM (Dayan et al., 2021)**   We provide detailed statistics of the partial EXAM dataset for each domain dimension in Tables 5 and 6. Site 7,8,9 are the three independent test sites that were not included during FL training. For the race groups, the "Other" category includes American Indian or Alaska Native, Native Hawaiian, or Other Pacific Islander, and patients with more than one race or unknown race. For the manufacturer dimension, the "Other" category includes Varian, Philips, Dongkang, Canon, Siemens, Samsung, Kodak, GE, and unknown manufacturer.

Table 5: Statistics of the race domain dimension and the label for the partial EXAM dataset used in our study.

| Site | White | Black | Asian | Latino | Other | $\geq$HFO % |
|------|-------|-------|-------|--------|-------|-------------|
| Site-1 | 59.6% | 10.0% | 3.4% | 2.0% | 24.9% | 12.4% |
| Site-2 | 75.0% | 11.1% | 2.8% | 0.6% | 10.5% | 9.1% |
| Site-3 | 46.5% | 26.3% | 4.2% | 7.0% | 16.0% | 9.6% |
| Site-4 | 71.4% | 6.3% | 4.2% | 0.8% | 17.2% | 11.4% |
| Site-5 | 44.0% | 28.4% | 1.6% | 6.3% | 19.8% | 9.9% |
| Site-6 | 0.0% | 0.0% | 100.0% | 0.0% | 0.0% | 18.8% |
| Site-7 | 75.7% | 7.1% | 1.1% | 0.0% | 16.2% | 2.4% |
| Site-8 | 82.9% | 2.8% | 2.1% | 0.2% | 12.0% | 11.0% |
| Site-9 | 79.1% | 7.9% | 1.2% | 2.4% | 9.4% | 2.4% |

Table 6: Statistics for the manufacturer and portable domain dimensions for the partial EXAM dataset used in our study.

| Site | Non-portable | Portable | Unknown | Agfa | Fuji | Konica Minolta | Thales | CareStream | Other |
|------|--------------|----------|---------|------|------|----------------|--------|------------|-------|
| Site-1 | 79.4% | 20.6% | 0.0% | 91.5% | 0.7% | 0.0% | 0.0% | 0.0% | 7.8% |
| Site-2 | 5.5% | 94.5% | 0.0% | 0.0% | 99.7% | 0.0% | 0.0% | 0.0% | 0.3% |
| Site-3 | 8.6% | 91.4% | 0.0% | 0.3% | 0.1% | 94.0% | 0.8% | 0.0% | 4.8% |
| Site-4 | 6.6% | 93.4% | 0.0% | 31.1% | 0.0% | 0.0% | 0.0% | 60.0% | 8.9% |
| Site-5 | 6.3% | 88.4% | 5.3% | 0.0% | 0.0% | 0.2% | 95.1% | 0.0% | 4.7% |
| Site-6 | 76.6% | 22.8% | 0.6% | 0.0% | 0.9% | 0.0% | 0.0% | 0.9% | 98.3% |
| Site-7 | 27.2% | 62.0% | 10.7% | 0.0% | 0.0% | 10.7% | 0.0% | 0.0% | 89.3% |
| Site-8 | 17.6% | 82.4% | 0.0% | 0.0% | 0.0% | 0.0% | 0.0% | 0.0% | 100.0% |
| Site-9 | 18.9% | 0.0% | 81.1% | 0.0% | 81.1% | 0.0% | 0.0% | 0.0% | 18.9% |

Table 7: Hyper-parameters for the domain shifts.

| Dataset | CMNIST | CelebA | FairFace | PovertyMap | Prostate MRI Seg. |
|---|---|---|---|---|---|
| Learning rate | 1e-3 | 1e-4 | 1e-4 | 1e-3 | 1e-4 |
| Weight decay | 1e-4 | 1e-4 | 1e-4 | 0 | 0 |
| Scheduler | n/a | n/a | n/a | ExponentialLR | ExponentialLR |
| Batch size | 32 | 32 | 32 | 64 | 16 |
| Type of mixup | mixup Zhang et al. (2017) | CutMix Yun et al. (2019) | CutMix | CutMix+C-Mixup Yao et al. (2022a) | n/a |
| Architecture | ResNet18 | ResNet50 | ResNet50 | ResNet18 | U-Net |
| Optimizer | SGD | SGD | SGD | Adam | Adam |
| Evaluation Iters | 200 | 100 | 100 | 200 | 6 |
| Pretrain Iters | 2,000 | 1,000 | 1,000 | 2,000 | 60 |
| Maximum Iters | 10,000 | 4,000 | 10,000 | 10,000 | 600 |

Table 8: Pearson correlation $r$ (higher is better) on out-of-distribution (unseen countries) held-out sets in PovertyMap-wilds. All results are averaged over 5 different OOD country folds, with standard deviations across different folds in parentheses.

|  | Validation (OOD) | Test (OOD) |
|---|---|---|
| **Average** | | |
| ERM | 0.80 (0.04) | 0.78 (0.04) |
| CORAL | 0.80 (0.04) | 0.78 (0.05) |
| IRM | 0.81 (0.05) | 0.77 (0.05) |
| Group DRO | 0.78 (0.05) | 0.75 (0.07) |
| C-Mixup | 0.81 (0.04) | **0.79** (0.05) |
| EMoE | 0.81 (0.03) | **0.79** (0.04) |
| **Worst** | | |
| ERM | 0.51 (0.06) | 0.45 (0.06) |
| CORAL | 0.52 (0.06) | 0.44 (0.06) |
| IRM | 0.53 (0.05) | 0.43 (0.07) |
| Group DRO | 0.46 (0.04) | 0.39 (0.06) |
| C-Mixup | 0.55 (0.07) | **0.50** (0.07) |
| EMoE | 0.55 (0.05) | **0.50** (0.07) |

## A.2 TRAINING DETAILS

All hyperparameters used across the datasets in the centralized setting are enumerated in Table 7. Hyperparameters that are common to all methods are consistently maintained. We have tuned all hyperparameters through a grid search approach. For mixup, it is applied only to sample pairs that share the same domain in at least one domain dimension or have the same label. All the models are trained with early stopping based on the performance on validation set.

For the federated learning setting, all the models are trained with $T = 20$ global communication rounds with Adam optimizer and a learning rate of $1 \times 10^{-4}$. For all methods we execute 3 epochs of update except FedDAR and FedEMoE , for which we do 3 epochs of head update and only 1 epoch of representation update. We set the hyperparameter for domain estimation loss as $\lambda = 0.1$.

## B ADDITIONAL EXPERIMENTS

### B.1 ADDITIONAL EXPERIMENTS ON SATELLITE IMAGE REGRESSION

**PovertyMap-Wilds (Koh et al., 2021)** This dataset consists of satellite images from 23 African countries, used for predicting the village-level real-valued asset wealth index. Each input is a 224 x 224 multispectral LandSat satellite image comprising eight channels, the label being the real-valued asset wealth index. The images are characterized by two domain dimensions: the countries and the urban/rural classification. The dataset includes 5 different cross-validation folds, with countries in these splits being disjoint to support an out-of-distribution setting. All experimental settings are in accordance with Koh et al. (2021).

**Results** All experiments were carried out using five distinct cross-validation folds. We report the Pearson correlation coefficient ($r$) for the sample average and the worst-case group performance in

Table 9: Worst group Pearson correlation r on PovertyMap-wilds with respect to different domain dimensions.

|  | Rural/Urban | Country | All |
|---|---|---|---|
| C-Mixup | 0.50 (0.07) | 0.72 (0.05) | 0.18 (0.14) |
| EMoE | 0.50 (0.07) | **0.75** (0.03) | **0.26** (0.06) |

Table 10: Dataset details including sample numbers and imaging protocols for each site in the MRI prostate segmentation task.

| Dataset | Institution | No. of Case | Field strength (T) | Resolution (in/through plane) (mm) | Endorectal Coil | Manufacturer |
|---|---|---|---|---|---|---|
| Site A | RUNMC | 30 | 3 | 0.6-0.625/3.6-4 | Surface | Siemens |
| Site B | BMC | 30 | 1.5 | 0.4/3 | Endorectal | Philips |
| Site C | HCRUDB | 19 | 3 | 0.67-0.79/1.25 | No | Siemens |
| Site D | UCL | 13 | 1.5 and 3 | 0.325-0.625/3.3-6 | No | Siemens |
| Site E | BIDMC | 12 | 3 | 0.25/2.2-3 | Endorectal | GE |
| Site F | HK | 12 | 1.5 | 0.625/3.6 | Endorectal | Siemens |

Table 8. The results demonstrate that the performance of EMoE is comparable to the previous state-of-the-art (SotA) method, C-mixup (Yao et al., 2022b). Further analysis presented in Table 9 reveals that EMoE method significantly outperforms C-mixup, exhibiting lower variance, particularly when considering the country dimension or both dimensions simultaneously. These findings highlight EMoE 's capability to account for all domain dimensions, thereby ensuring fairer performance across them. They also confirm the adaptability of EMoE for application in image regression tasks.

### B.2   ADDITIONAL EXPERIMENTS ON MEDICAL IMAGE SEGMENTATION

**Dataset Details**   We utilize a multi-site dataset for prostate MRI segmentation as introduced by Liu et al. (2020). This dataset comprises prostate T2-weighted MRI data, complete with segmentation masks, gathered from six distinct data sources spanning three public datasets. The sample counts and imaging protocols for each site are summarized in Table 10. We adopt three domain dimensions including field strength, endorectal coil, and manufacturers. For site D, the specific field strength for each case is not labeled, so we treat all of them as unknown. For segmentation, 2D patches are extracted from the 3D volumes. Each sample is resized to 384x384 pixels in the axial plane and normalized to zero mean and unit variance, in accordance with the preprocessing steps described by Liu et al. (2020).

**Implementation Details**   In line with the leave-one-domain-out strategy detailed in Liu et al. (2020), we train on data from $K - 1$ sites while testing on the remaining unseen target site. We employ a U-Net (Ronneberger et al., 2015) architecture enhanced with dilated convolution, comprising four blocks in both the encoder and decoder, and a bottleneck depth of four. The output feature dimension of the first layer is set to 32. Other detailed hyperparameters are enumerated in Table 7.

**Results**   As depicted in Table 11, EMoE markedly enhances out-of-distribution generalization, particularly for Sites B, C, and E, where ERM's performance suffers due to a significant domain gap from the source sites. These results corroborate the adaptability of EMoE method to more complex tasks such as Medical Image Segmentation.

Table 11: Results on MRI prostate segmentation with different sites as held-out test set. Dice (%) score are reported.

| Method | Site A | Site B | Site C | Site D | Site E | Site F | Average |
|---|---|---|---|---|---|---|---|
| ERM | 91.8 | 58.9 | 55.1 | 84.8 | 69.5 | 85.4 | 74.3 |
| EMoE (Avg) | **92.9** | **65.0** | 58.8 | 84.0 | 72.1 | **88.0** | **76.8** |
| EMoE (Max) | 92.3 | 62.7 | 60.8 | 81.2 | 72.9 | 85.8 | 76.0 |
| EMoE (Pro) | 91.8 | 56.6 | **61.5** | **86.4** | **73.6** | 86.7 | 76.1 |

## B.3 ABLATION STUDY

We conducted ablation experiments with different configurations of EMoE on the CelebA and Fair-Face datasets to evaluate the effectiveness of each component of our method. The results are presented in Table 12. All the results use probabilistic aggregation.

Table 12: The worst group accuracy across all domain dimensions for different configurations of EMoE. All results are averaged over 3 runs with different random seeds, with standard deviations in parentheses.

| Estimation | Pretrain | CelebA | FairFace |
|---|---|---|---|
| Hard | No | $77.4 \pm 2.3\%$ | $70.3 \pm 3.7\%$ |
| GT | Only Pretrain | $75.8 \pm 1.3\%$ | $\mathbf{75.6 \pm 1.6\%}$ |
| Soft | Yes | $82.3 \pm 0.5\%$ | $71.8 \pm 2.5\%$ |
| Hard | Yes | $\mathbf{83.9 \pm 1.3\%}$ | $75.1 \pm 1.6\%$ |

Table 13: The worst group accuracy across all domain dimensions when EMoE take different dimensions of domains into account. All results are averaged over 3 runs with different random seeds, with standard deviations in parentheses.

| Domain Dim. | CelebA | FairFace |
|---|---|---|
| dim1 | $80.6 \pm 1.3\%$ | $57.4 \pm 6.1\%$ |
| dim2 | $54.8 \pm 1.3\%$ | $65.1 \pm 5.0\%$ |
| dim1 x dim2 | $80.6 \pm 0.6\%$ | $58.1 \pm 1.9\%$ |
| EMoE | $\mathbf{83.9 \pm 1.3\%}$ | $\mathbf{75.1 \pm 1.6\%}$ |

**Effects of Pretraining on EMoE**  A comparison between the first and fourth rows of Table 12 demonstrates that proper pretraining significantly improves the initialization for the domain estimator and domain-specific predictor, enabling the model to achieve enhanced performance.

**Effects of Different Domain Estimation Strategies**  An analysis of the last three rows of Table 12 reveals that a hard domain assignment for the domain-wise personalized predictor surpasses the performance of soft assignment. This suggests that outputs for different domains cannot be effectively combined in a simple linear fashion. Employing the actual domain label (GT) surpasses hard assignment on the FairFace dataset, likely because certain domain information is not readily inferable from the image alone; incorporating the actual domain label provides additional context. We adopt hard assignment as our standard approach, operating under the assumption that domain labels are unavailable at test time.

**Effectiveness of MoEs Ensemble Design**  The effectiveness of the MoEs ensemble design is evaluated under various settings. The results are shown in Table 13. For Dim.1 or Dim.2, a single MoE is employed, focused exclusively on the corresponding domain dimension. In contrast, Dim.1 x Dim.2 considers both dimensions, yet it treats the combined domain uniformly with a single MoE as well. The results demonstrate that the ensemble of MoEs specialized for all domain dimensions substantially enhances the performance for the worst-case group considering all dimensions. Moreover, the MoEs ensemble approach, which capitalizes on the inherent structure of domain information, proves superior to a uniform treatment of multi-dimensional domains.

## B.4 ADDITIONAL RESULTS

The full results for centralized learning evaluation, inclusive of standard errors from trials with three distinct random seeds, are presented in Table 14.

Table 14: Full results of centralized learning evaluation with standard error. We show the worst group accuracy regarding only one or all domain dimensions.

| | | CMNIST | | | CelebA | | | FairFace | | |
|---|---|---|---|---|---|---|---|---|---|---|
| | | dim1 | dim2 | all | dim1 | dim2 | all | dim1 | dim2 | all |
| ERM | | $97.4 \pm 0.2\%$ | $97.6 \pm 0.0\%$ | $96.0 \pm 0.3\%$ | $79.1 \pm 0.2\%$ | $90.3 \pm 0.3\%$ | $25.8 \pm 1.3\%$ | $80.4 \pm 0.1\%$ | $80.3 \pm 0.4\%$ | $45.0 \pm 1.7\%$ |
| UW | | $97.4 \pm 0.1\%$ | $97.8 \pm 0.0\%$ | $96.8 \pm 0.1\%$ | $90.2 \pm 0.1\%$ | $92.5 \pm 0.2\%$ | $73.1 \pm 2.2\%$ | $83.4 \pm 0.5\%$ | $83.6 \pm 0.6\%$ | $72.0 \pm 2.1\%$ |
| RIDG | | $98.2 \pm 0.1\%$ | $98.4 \pm 0.0\%$ | $97.2 \pm 0.1\%$ | $90.2 \pm 0.1\%$ | $92.6 \pm 0.1\%$ | $71.5 \pm 1.9\%$ | $84.1 \pm 0.1\%$ | $84.6 \pm 0.3\%$ | $73.1 \pm 1.6\%$ |
| SAGM | | $98.3 \pm 0.1\%$ | $98.3 \pm 0.0\%$ | $97.3 \pm 0.1\%$ | $90.8 \pm 0.5\%$ | $92.8 \pm 0.1\%$ | $80.1 \pm 0.5\%$ | $84.5 \pm 0.5\%$ | $84.3 \pm 0.6\%$ | $74.1 \pm 1.0\%$ |
| IRM | dim1 | $97.6 \pm 0.1\%$ | $97.9 \pm 0.1\%$ | $96.3 \pm 0.1\%$ | $90.7 \pm 0.1\%$ | $92.6 \pm 0.1\%$ | $72.6 \pm 1.3\%$ | $83.6 \pm 0.2\%$ | $83.8 \pm 0.2\%$ | $72.1 \pm 1.0\%$ |
| | dim2 | $97.9 \pm 0.1\%$ | $97.9 \pm 0.1\%$ | $96.6 \pm 0.1\%$ | $88.6 \pm 0.1\%$ | $93.1 \pm 0.1\%$ | $63.4 \pm 1.2\%$ | $83.2 \pm 0.1\%$ | $83.7 \pm 0.1\%$ | $66.4 \pm 2.6\%$ |
| | all | $97.8 \pm 0.1\%$ | $97.9 \pm 0.1\%$ | $97.0 \pm 0.2\%$ | $90.3 \pm 0.2\%$ | $92.5 \pm 0.1\%$ | $72.6 \pm 2.7\%$ | $83.0 \pm 0.1\%$ | $83.1 \pm 0.2\%$ | $70.2 \pm 1.3\%$ |
| LISA | dim1 | $97.5 \pm 0.1\%$ | $97.8 \pm 0.1\%$ | $96.4 \pm 0.3\%$ | $91.1 \pm 0.2\%$ | $92.2 \pm 0.1\%$ | $80.0 \pm 0.4\%$ | $82.9 \pm 0.1\%$ | $82.7 \pm 0.2\%$ | $55.8 \pm 2.9\%$ |
| | dim2 | $97.6 \pm 0.1\%$ | $97.5 \pm 0.1\%$ | $96.2 \pm 0.3\%$ | $86.4 \pm 0.2\%$ | $92.5 \pm 0.1\%$ | $50.5 \pm 1.7\%$ | $82.4 \pm 0.2\%$ | $83.1 \pm 0.1\%$ | $61.6 \pm 3.3\%$ |
| | all | $98.0 \pm 0.1\%$ | $97.9 \pm 0.1\%$ | $96.9 \pm 0.1\%$ | $90.8 \pm 0.3\%$ | $92.2 \pm 0.1\%$ | $77.4 \pm 1.5\%$ | $82.5 \pm 0.1\%$ | $82.6 \pm 0.1\%$ | $71.2 \pm 0.6\%$ |
| GroupDRO | dim1 | $97.4 \pm 0.2\%$ | $97.9 \pm 0.1\%$ | $96.6 \pm 0.1\%$ | $90.7 \pm 0.3\%$ | $92.3 \pm 0.1\%$ | $73.1 \pm 0.9\%$ | $83.3 \pm 0.2\%$ | $83.0 \pm 0.3\%$ | $73.0 \pm 0.4\%$ |
| | dim2 | $98.0 \pm 0.1\%$ | $97.9 \pm 0.1\%$ | $96.6 \pm 0.2\%$ | $87.8 \pm 0.2\%$ | $93.0 \pm 0.1\%$ | $59.2 \pm 0.5\%$ | $83.2 \pm 0.1\%$ | $83.6 \pm 0.2\%$ | $67.4 \pm 1.2\%$ |
| | all | $97.9 \pm 0.1\%$ | $98.0 \pm 0.1\%$ | $96.5 \pm 0.2\%$ | $89.7 \pm 0.2\%$ | $91.8 \pm 0.1\%$ | $76.9 \pm 0.5\%$ | $82.5 \pm 0.2\%$ | $82.9 \pm 0.3\%$ | $73.7 \pm 1.2\%$ |
| **EMoE (ours)** | AVG | $98.5 \pm 0.0\%$ | $98.3 \pm 0.1\%$ | $97.4 \pm 0.1\%$ | $91.5 \pm 0.2\%$ | $92.2 \pm 0.1\%$ | $81.1 \pm 1.7\%$ | $82.7 \pm 0.2\%$ | $83.2 \pm 0.1\%$ | $71.8 \pm 0.8\%$ |
| | MAX | $98.2 \pm 0.0\%$ | $98.3 \pm 0.0\%$ | $\mathbf{97.5 \pm 0.1\%}$ | $91.4 \pm 0.1\%$ | $92.2 \pm 0.1\%$ | $82.3 \pm 0.8\%$ | $83.1 \pm 0.3\%$ | $83.7 \pm 0.1\%$ | $72.1 \pm 1.1\%$ |
| | PRO | $98.4 \pm 0.1\%$ | $98.3 \pm 0.1\%$ | $97.1 \pm 0.3\%$ | $92.1 \pm 0.1\%$ | $92.5 \pm 0.1\%$ | $\mathbf{83.9 \pm 0.8\%}$ | $83.9 \pm 0.2\%$ | $83.8 \pm 0.3\%$ | $\mathbf{75.1 \pm 0.9\%}$ |

## C    Methodology

**Probabilistic Aggregation for Regression**    In Section 3.3, we introduced Probabilistic Aggregation in the context of classification. Here, we demonstrate that Equation 8 is also applicable to regression models. Recall that $o_k(x)$ represents the prediction of the $k$-th MoE, implying a distribution $p(y|x, z_k) = \mathcal{N}(o_k(x), \sigma^2)$, or equivalently, $p(y|x, z_k) \propto \exp(-\frac{1}{2\sigma^2}(y - o_k)^2)$. Therefore, based on Equation 7, we have $p(y|x, z_1, \cdots, z_k) \propto \exp\left(-\frac{1}{2\sigma^2}\left(\sum_{k=1}^{K}(y - o_k)^2\right) - (K-1)(y - o_{\texttt{base}})^2\right)$. By maximizing the joint distribution, we obtain $o = \arg\max_y p(y|x, z_1, \cdots, z_k) = \sum_{k=1}^{K} o_k - (K-1)o_{\texttt{base}}$, aligning with the formula in Equation 8.

