# OpenReview forum: "Learning Under Multi-dimensional Domain Shifts: A Ensemble of Mixtures of Experts Approach"
_ICLR.cc/2025/Conference — Submitted to ICLR 2025_

### Official Review · Reviewer_5PRQ · 2024-10-31

**Soundness:** 2
**Presentation:** 2
**Contribution:** 2
**Rating:** 3
**Confidence:** 5

**Summary:**

This paper addresses the problem of multi-dimensional domain shifts in both centralized and federated learning scenarios. Traditional methods often treat domain shifts as occurring along a single dimension, overlooking the complexity of real-world data, where shifts occur across multiple factors, such as demographics, equipment types, or protocols. The proposed approach, Ensemble of Mixtures of Experts (EMoE), seeks to address this complexity by leveraging multiple Mixtures of Experts (MoEs), each specialized for a specific domain dimension.

**Strengths:**

1. The focus on multi-dimensional domain shifts represents a novel perspective in domain adaptation research. Unlike most traditional approaches that treat domain shifts as a single homogeneous shift, the paper introduces the concept of multi-dimensional shifts which reflect the complexities of real-world data (e.g., shifts in demographics, equipment types, and geographic regions). This novel framing addresses a significant gap in existing domain generalization literature.

2. Addressing both centralized and federated learning under a unified framework adds to the originality of the work. The proposed framework is not only capable of adapting to centralized scenarios but is also extended to federated settings with a version called FedEMoE.

**Weaknesses:**

1. The problem definition of multi-dimensional domain shifts is not sufficiently explained, and the significance of studying these shifts in both centralized and federated learning contexts is unclear. The current motivation does not adequately answer the question of why addressing multi-dimensional domain shifts is crucial and what specific value is added by tackling centralized and federated learning scenarios simultaneously.

2. The novelty of the EMoE approach is limited, as it essentially builds upon existing Mixture of Experts (MoE) methods. While combining multiple MoEs is an interesting idea, it does not provide substantial innovation beyond what is already known in the literature.

3. The experimental evaluation is limited to only three simple datasets. These datasets may not capture the complexities involved in real-world applications of multi-dimensional domain shifts, which raises concerns about the generalizability of the proposed approach.

4. The comparison with existing methods is incomplete, as several relevant and recent studies are not considered. This omission weakens the validity of the claims regarding the superiority of the proposed method.

**Questions:**

1. Why is it important to study multi-dimensional domain shifts in real-world scenarios? What practical problems are you solving that current domain shift approaches do not effectively address?

2. What is the novel theoretical or technical contribution of the EMoE model beyond simply employing multiple MoEs? How does EMoE advance beyond the standard use of MoEs in handling domain shifts?

3. Why did you choose to evaluate on these specific datasets? How representative are they of real-world scenarios that involve multi-dimensional domain shifts?

4. Why were some recent and relevant works not included in the comparisons? For instance, "An Iterative Self-Learning Framework for Medical Domain Generalization" (NeurIPS 2024) should be highly relevant, as it addresses domain generalization in the muliple dimensions domains.

---

### Official Review · Reviewer_PxG9 · 2024-11-01

**Soundness:** 2
**Presentation:** 1
**Contribution:** 2
**Rating:** 3
**Confidence:** 4

**Summary:**

This paper considers the training of neural network under datasets domain shift. Unlike traditional approaches that typically consider a single factor causing domain shift, the authors investigate multiple factors. They propose using a mixture of experts model for each factor individually and then combine these mixtures to handle multiple factors collectively. Additionally, they design a 'domain estimator' to account for scenarios where factor labels may be missing in dataset.

**Strengths:**

- **Setting.** Viewing domain shift as a result of multiple factors is new.
- **Various settings.** The authors explore a range of settings, from centralized to federated frameworks, as well as cases where domain labels are fully available or partially missing. However, this breadth of scenarios could be a potential weakness, as some extensions appear relatively straightforward and detract from the clarity of critical content, especially within the constraints of the page limit.

**Weaknesses:**

- **Method formulation.** The multiple-factor domain shift problem could be simplified to a single-factor domain shift by introducing a single factor with $M^K$ levels. However, the authors do not discuss this possibility, which leaves the necessity and advantages of the proposed approach less compelling.

- **Unclear writing.** The paper lacks clarity of key sections, as detailed below:
    - L270–L276: The phrase "inspired by the above equation, xxx" lacks clarity, as the connection between equations (3) and (4) is not explained.
    - Section 3.4: While a "domain estimator" is introduced, neither its input nor output is clearly described, making the section challenging to follow.
   - L304: The notation $\bf{l}_k$ and $\tilde{\bf{z}}_k$ are used without definition, leading to ambiguity.
   - Training Process Description: The section detailing the training process is vague, with statements in L316-317 being particularly unclear for the reader.
   - Handling Missing Domain Labels: Although handling missing domain labels is presented as a novel aspect, the explanation lacks clarity, leaving the implementation of this approach ambiguous.
- **Incomplete comparison with baselines.**  In the experimental results (Table 4), the paper does not compare with personalized federated learning (PFL) methods discussed in the related work, which would provide a more comprehensive evaluation.

**Questions:**

**Method**
1. In the proposed approach, each factor is represented by a mixture of experts (MoE), where each level corresponds to a component in the mixture. This raises the question: why is an additional domain estimator necessary, rather than using the MoE output directly to estimate the domain?
2. Given the unclear description of the domain estimator, it is unclear how the authors handle missing labels. Specifically, without complete labels, how can they determine the number of factors and the number of levels for each factor?
3. How about directly encoding the labels as a feature as concate it to the input and use base methods?

**Experiments**
1. Why using worst group accuracy as the metric? How do these methods differ if we use average accuracy as the performance metric

The reviewer would suggest the authors to only focusing on the centralized setting and describe your methods more carefully and clearly.

---

### Official Review · Reviewer_Mz6q · 2024-11-03

**Soundness:** 2
**Presentation:** 2
**Contribution:** 3
**Rating:** 5
**Confidence:** 4

**Summary:**

The paper addresses the complex issue of domain shifts in deep learning, especially under scenarios where these shifts occur simultaneously across multiple dimensions, such as different demographics, equipment types, and protocols. This work introduces the Ensemble of Mixtures of Experts (EMoE) approach and its federated learning version, FedEMoE. The main contribution lies in tackling domain shifts with an ensemble of specialized models, each targeting a specific dimension of domain variation. A key innovation is the integration of a domain estimator capable of functioning effectively even when domain labels are incomplete or unreliable.

**Strengths:**

1. The paper presents a strategy for handling multi-dimensional domain shifts, a significant step beyond traditional domain generalization methods that often consider only one dimension at a time.

2.  The method applies to both centralized and federated learning settings, which is particularly valuable for real-world applications involving heterogeneous and decentralized data sources

3. Including a domain estimator that can bypass reliance on domain labels at inference time demonstrates practical foresight for real-world data where domain information may not always be complete or reliable.

**Weaknesses:**

1. Did not understand Table 2 fully. There are dim1 and dim2 in rows vs colums. What do they represent for each dataset? For example, celebA biases can be due to gender, smile, etc. (all attribute labels except blond vs. non blond). Does the author refer to compute WGA for each of such group? Then (dim1,dim1) (row vs col) will be WGA b/w them, which does not make sense. (dim1, dim2) will be the WGA of that group. Is that what they mean? They should write it clearly.


2. The mechanism of how the domain estimator adapts during inference when no labels are provided needs more detail. The paper describes the use of Gumbel-Softmax and one-hot encoding for training, but the explanation of how robust these techniques are to noisy data or real-world discrepancies is insufficient. The authors can use Huber loss or uncertainty estimation techniques because this part needs careful consideration for scenarios where no domains are labeled. So, different potential options have to be evaluated.


3. The domain estimator's training details are briefly explained but not explored in depth. For example, the paper mentions using cross-entropy loss for direct supervision when domain labels are available. However, the performance trade-offs between predefined and estimated labels in noisy settings are not fully analyzed. I want to see a table/plot where results will be given when domain labels are used and not used in the method to see the trade-off.


4. The use of multiple MoEs and an aggregation framework inherently adds computational overhead. Did they perform any computational complexity analysis? Popular methods like DFR, JTT, and GroupDRO are not this computationally expensive.

5. Regarding baselines, two important baselines are missing—DFR (ICLR 2022) and Whac-A-Mole (CVPR 2023). DFR uses balanced validation data to learn robust representation. So, I want to see how MoE performs compared to DFR. Also, MoE tackles multiple domain shifts. All baselines in the paper do not encounter that. So, a baseline method like Whac-A-Mole needs to be compared against, as Whac-A-Mole tackles the problem of multiple biases.

6. When no domain label exists, the author proposed a smart domain estimator. So, a detailed experiment is needed on how well this domain estimator performs. Seeing only WGA won't help. They can look into slice discovery literature (e.g, DOMINO (ICLR 2022), FACTS (ICCV 2023)). Each domain can be considered a slice or even a source of bias. They can be compared with slice discovery methods or even with the ground truth domains.

7. For centralized experiments, the datasets are too toyish. The authors can use popular chest-x-ray  (CXR)Datasets or NLI datasets. See Subpopshift paper (Yuzhe Yang et el. ICML 2023). For ex, pneumothorax classification in CXR datasets (NIH-CXR, MIMIC-CXR, CheXpert) contains domain shift due to Chesttubes. Recently, LADDER: Language Driven Slice Discovery and Error Rectification has used a lot of medical imaging datasets, from CXR to Mammograms, which can be used to evaluate rigorously.


8. The captions of tables are too short and need elaborated descriptions.

**Questions:**

see weakness

---

### Official Review · Reviewer_5hZi · 2024-11-04

**Soundness:** 3
**Presentation:** 3
**Contribution:** 1
**Rating:** 3
**Confidence:** 4

**Summary:**

This paper attempts to show that sample re-weighting or data balancing when combined with mixture of experts architecture (where experts are provided explicit domain labels) can be effective for addressing multi-dimensional domain shift a.k.a subpopulation shift.

The paper combines the idea of using MOE for domain generalization which is known in the literature [1] and data balancing / reweighting which is also well studied [2].

[1] Li, Bo, et al. "Sparse mixture-of-experts are domain generalizable learners.
[2] Idrissi, Badr Youbi, et al. "Simple data balancing achieves competitive worst-group-accuracy."

Empirically, the proposed method performs better than some standard baselines like ERM, IRM, GroupDRO etc. The method was also extended to federated learning where sample sizes may differ across clients. In such setting the re-weighting scheme needs to account for the effective size of the subgroup within each client.

Overall, the paper presents fairly sound ideas to a very important research area with strong empirical results but doesn't contribute much new knowledge to the study of either domain generalisation, MOEs, or federated learning. ]

The technical contribution regarding re-weighting is to incorporate the size of the sub-domain into the scaling factor. This type of importance weighting is a well-known technique, but it can suffer from overfitting, an issue the paper does not address. A key component of the paper is to enable the MOEs to learn explicit domain information, which may not be trivial, as it is generally unclear whether MOEs can do this implicitly. The technical contribution in this regard is introducing a non-differentiable one-hot encoding; however, the paper does not provide any evidence that this approach has any effect.

**Strengths:**

Subpopulation shift in federated learning is less studied and I appreciate this paper for providing a setup for the problem and rigorously evaluating their method within this context.

**Weaknesses:**

The main weakness is really in the contribution. Perhaps, it could be said that the area for new contribution is in federated learning but the paper needs to make federated learning the main problem setup, and contextualize the contributions with what is already known e.g. See [1] which studies settings where the shifting "dimension" or "domain" is "space" and "time" across clients.

[1] Jothimurugesan, Ellango, et al. "Federated learning under distributed concept drift."

**Questions:**

Are the baseline (i.e ERM, IRM, etc.) also trained with the same MOE architecture?

---

### Official Review · Reviewer_uvdw · 2024-11-04

**Soundness:** 3
**Presentation:** 4
**Contribution:** 3
**Rating:** 6
**Confidence:** 3

**Summary:**

The authors present a comprehensive study on addressing distribution shift across multiple dimensions simultaneously that they apply to both centralised and federated setups. To this end they introduce an ensemble of Mixture of Experts (MoE), with each MoE specialized for a different domain dimension.

**Strengths:**

Excellent presentation and comprehensive evaluation both in the federated and centralised setting; very ambitious experiments on real-world shifts.

**Weaknesses:**

Since the authors aim to benchmark their method both in the centralised and federated setting, in each setup experiments are a bit limited. For example, I find the experiments on evaluating the effect on missing domain labels limited and would like to see results on the more complex EXAM and FairFace datasets.

Additionally, I am concerned about the computational complexity of the MoE approach. How does this compare to baselines

Having read the other reviewers' concerns, I adjust my rating.

**Questions:**

See above.

---

### Meta-Review · Area_Chair_XcA4 · 2024-12-21

**Metareview:**

The paper addresses the  issue of domain shifts in deep learning, under scenarios where
these shifts occur simultaneously across multiple dimensions.

Reviewers raised several concerns about clarity of the paper, clarity of the contributions, as well
as on the experiments, particularly about baselines related to personalized federated learning.
as such, they are not keen to accept the paper in its current status.

**Additional Comments On Reviewer Discussion:**

authors did not provide rebuttals.

---

### Decision · Program_Chairs · 2025-01-22

Reject